# A Fluorescent Tracer Based on Castor Oil for Monitoring the Mass Transfer of Fatliquoring Agent in Leather

**DOI:** 10.3390/ma15031167

**Published:** 2022-02-02

**Authors:** Hongmei Wen, Yulu Wang, Hongxia Zhu, Liqiang Jin, Feifei Zhang

**Affiliations:** Faculty of Light Industry, Qilu University of Technology (Shandong Academy of Sciences), Jinan 250353, China; W15610143101@163.com (H.W.); hxzhu@qlu.edu.cn (H.Z.); fei626-918@163.com (F.Z.)

**Keywords:** fluorescent tracing technique, castor oil, fatliquoring agent, mass transfer, leather

## Abstract

Fatliquoring is one of the most important processes in leather making, in which the collagen fibers are split effectively, while the crust acquires a good softness and mechanical strength. The effectiveness of fatliquoring depends not only on the variety and dosage of fatliquoring agent but also on its distribution in hierarchical collagen fiber layers. Up to now, the research about the mass transfer of fatliquoring agent in leather is very limited because quantifying the distribution of invisible fatliquor in leather is very difficult. In this paper, a fluorescent tracing technique based on modified castor oil was established and send used to investigate the mass transfer of fatliquor in leather. The crucial fluorescent tracer was synthesized favorably by the reaction of castor oil, successively with maleic anhydride and 5-aminofluorescein, which was confirmed by FTIR, ^1^H NMR and DSC. The fluorescent tracer was pH-sensitive and emitted an intense fluorescent signal at pH 8–10. Then, it was applied to monitor the fatliquoring process in real-time. Compared with Sudan IV or Nile Blue sulphate dyeing tests, this fluorescent tracing technique could conveniently visualize and quantify the penetration and distribution of fatliquor in leather process.

## 1. Introduction

Fatliquoring is an indispensable process in leather manufacturing that plays a key role in giving the leather satisfactory flexibility and keeping its mechanical properties [1,2]. Generally, degreasing is necessary before tanning in order to be beneficial for the penetration and distribution of the tanning agent, from which the natural fats of skins or hides are removed. Without the lubrication of natural fats, the tanned collagen fibers will stick together when the water loss and cracking of wet blue takes place easily [3]. Therefore, the leather must be fat-liquored using the emulsions of animal fats, vegetable oils, synthetic oils or their derivatives before drying [4,5]. After being fat-liquored, the split collagen fibers of re-tanned leather are further enveloped by very thin oil membranes. Correspondingly, the mechanical and physical properties and water-repellent performances of leather are improved.

It should be noted that the improvement of the fatliquoring agent in the mechanical properties of leather is strongly associated with its distribution in the crust. As a special natural protein matrix, leather is different from the textile fabrics of cotton and linen. The thickness, compactness and random-woven property of leather have a significant effect on the penetration and distribution of the fatliquoring agent and other leather chemicals [6]. Additionally, the chemicals previously combined with collagen fibers will influence the penetration and combination of subsequent chemicals. A typical example is the fading effect of acrylic polymers on dye. It is clear that the penetration and distribution of the fatliquoring agent in the whole cross section of leather is a very complex process.

Therefore, the investigation of the mass-transfer processes of the fatliquoring agent in wet blue, as well as its distribution, are crucial to understanding its structure–activity relationship, which is very important for designing and developing new fatliquoring agents. However, the research in this field is mainly focused on developing new kinds of fatliquoring agents and rarely focuses on their mass transfer and distribution in wet blue. It is limited to the shortage of effective and real-time tracing technique for fatliquoring agents. Scanning electron microscope-energy dispersive X-ray (SEM-EDX) analysis is a semi quantitative method for mapping the composition of metal elements such as chrome and aluminum in leather samples [7]. Unfortunately, this method is not suitable for the analysis of organic fatliquoring agents, mainly composed of carbon, hydrogen and oxygen. Sudan IV staining test or Nile Blue sulphate dyeing test are traditionally used to investigate the distribution of oil after fatliquoring. However, these staining methods are relatively complicated in sample preparation, and the slice is easily broken in the dyeing process. More importantly, these methods are difficult in monitoring the mass transfer of fatliquoring agents in real-time. Consequently, the construction of a real-time tracing technique for monitoring the mass transfer of fatliquoring agent in leather is imperative.

The fluorescent tracing technique is an effective and reliable analytical method with a high sensitivity in detecting target material [8,9,10,11,12]. With the help of the fluorescent tracing technique, fortunately, the mass transfer and distribution of leather chemicals or proteins/enzymes can be researched visually and quantitatively. Yang Qian et al., used the bovine serum albumin labeled with fluorescein isothiocyanate as an enzyme model to investigate the transfer of protein materials in hides, which confirmed that the fluorescent tracing technique was suitable for revealing the mass transfer and action mechanism of enzymes in leather making [13]. Jinxia Du et al. synthesized a fluorescent tracer by introducing 3-acryloyl fluorescein into the poly (acrylic-co-stearyl acrylate) by radical copolymerization to monitor the penetration of amphiphilic acrylate copolymer in real time, and this tracer could effectively reflect the penetration depth of amphiphilic copolymers in leather [6]. Yunhang Zeng et al. prepared a 5-aminofluorescein-labeled poly (acrylic acid) and applied it in the re-tanning process in order to study the mass transfer of acrylic resins and to discover their action mechanism [14]. This research proved the feasibility and effectiveness of the fluorescent tracing technique in monitoring the mass transfer of leather chemicals. However, there are few papers that investigated the mass transfer of fatliquoring agents in leather by using the fluorescent tracing method.

Castor oil, as one of the few vegetable oils naturally containing hydroxyl groups, is a renewable raw material widely used in the preparation of bio-based polymers, such as polyesters, polyurethanes, adhesives, and so on [15,16,17]. In fact, castor oil is the starting material for one kind of traditional leather fatliquoring agent, namely sulfated castor oil, commonly known as Turkey Red oil. Given its good emulsibility in water and lubrication of collagen fibers, sulfated castor oil is an ideal model substance for investigating the mass transfer of fatliquoring in chrome-tanned leather, while the important aspect of researching its mass transfer is to develop a fluorescent tracer that has a similar structure with sulfated castor oil. Obviously, synthesizing a tracer directly derived from castor oil is feasible and effective.

In this paper, a fluorescent tracing technique based on castor oil was established for investigating the mass transfer and distribution of fatliquor in leather. Firstly, carboxylate castor oil (CCO) was prepared by the reaction between castor oil and maleic anhydride. Then, a fluorescent tracer was synthesized by the reaction of CCO with 5-aminofluorescein (AF). Finally, the mass transfer of sulfated castor oil containing 0.5% fluorescent tracer in goat wet blue was investigated in detail.

## 2. Materials and Methods

### 2.1. Materials

Analytically pure castor oil (saponification value: 180 mgKOH/g), maleic anhydride, dimethylformamide, butoxyethanol and chemical pure sulfated castor oil (Turkey Red Oil sodium salt, 60%) were supplied by Sinopharm Chemical Reagent Co., Ltd. (Shanghai, China). The following chemicals: 1-(3-Dimethylaminopropyl)-3-ethylcarbodiimide hydrochloride (EDC), 5-aminofluorescein (AF), Sudan IV and Nile Blue sulphate, were purchased from Macklin Biochemical Co., Ltd. (Shanghai, China). Goat skin in a wet blue state was obtained from Shandong Juncheng Leather Co., Ltd. (Linyi, China). A technical-grade basic chromium sulphate (CTS, basicity, 33%; Cr_2_O_3_ content, 25%) was supplied by Brother Enterprise Holding Co., Ltd. (Zhejiang, China), and a technical grade non-ionic degreasing agent (FB, 40%) was offered by Leahou Light Industrial of New Material Co., Ltd. (Dezhou, China). Formic acid, sodium formate, acetic acid, sodium carbonate, dichloromethane, ethyl alcohol, acetone, hydrochloric acid and sodium hydroxide were analytical reagents provided by Tianjin Fuyu Fine Chemical Co., Ltd. (Tianjin, China).

### 2.2. Preparation of Fluorescent Fatliquoring Agent

#### 2.2.1. Preparation of the Fluorescent Tracer

The preparation procedures of the fluorescently labeled castor oil by AF are illustrated in Figure 1. Firstly, 93.5 g of castor oil was added to a four-neck flask equipped with a mechanical stirrer and heated in an oil bath. Then, 19.6 g of maleic anhydride was introduced into the system at 90 °C and reacted for 4 h to obtain carboxylate castor oil (CCO). Subsequently, 113.1 mg CCO was dissolved in 20 mL dimethylformamide on a magnetic stirring apparatus at room temperature. After the carboxylate castor oil was dissolved completely, 34.7 mg AF was reacted with the above modified oil for 60 min under the catalyzation of 19.1 mg EDC. After that, the product was mixed evenly with 30 mL dichloromethane. Several hours later, the dichloromethane layer was collected and remixed evenly with 90 mL distilled water. When the organic phase separated thoroughly from the water phase, the water phase was discharged to remove the EDC. The washing process of the organic phase with distilled water was repeated three times. Finally, the solvent of the organic phase was removed by using rotary evaporators and the obtained product was a fluorescently labeled tracer (AFCO).

#### 2.2.2. Preparation of Fluorescent Fatliquoring Agent

An amount of 100 mg of AFCO dissolved in 0.5 mL butoxyethanol was mixed with 20 g sulfated castor oil on Vortex-Genie 2 (Scientific Industries, Bohemia, NY, USA) for 1 min to prepare a fluorescent fatliquoring agent, which was named AFSCO and used as common anionic fatliquoring agent.

### 2.3. Characterization

#### 2.3.1. Fourier Transform Infrared (FTIR) Spectra

The dried castor oils, CCO and AFCO were dropped on the BaF_2_ wafer for FTIR spectra analysis, respectively. Their infrared spectra were obtained by TENSOR-27 (Bruker, Ettlingen, Germany) over a range of 500–4000 cm^−1^ with a nominal resolution of 2 cm^−1^ at room temperature.

#### 2.3.2. Nuclear Magnetic Resonance (NMR) Spectrometry

About 15 mg of castor oil, CCO and AFCO, respectively, were dissolved in 0.5 mL of CDCl_3_ solvent in a 5 mm diameter sample tube for NMR of ^1^H (AVANCE II 400, Bruker, Billerica (MA), Germany) analysis.

#### 2.3.3. Differential Scanning Calorimetry (DSC)

About 10 mg of castor oil, CCO and AFCO, respectively, were sealed in an aluminum pan with an empty, sealed aluminum pan as the reference. The DSC curves of the above oil samples were conducted on DSC250 (TA Instruments, Milford, MA, USA) in the temperature range from −80 °C to 80 °C with a constant heating rate of 5 °C/min under nitrogen flow.

#### 2.3.4. Fluorescence Emission Spectra

The AFCO was dissolved in a 75% (*v*/*v*) ethanol solution to a concentration of 50 mg/L, whose pH was adjusted by using 0.1 mol/L HCl and 0.1 mol/L NaOH to 6, 7, 8, 9 and 10, respectively. The fluorescence emission spectra of the solutions above were measured with an excitation wavelength of 486 nm using a fluorescence spectrophotometer (Cary Eclipse, Agilent, Santa Clara (Cal.), CA, USA).

### 2.4. Establishment of Fluorescent Tracing Technique of Fatliquoring Agent

#### 2.4.1. The Fatliquoring Process of Goat Skin Wet Blue

The fatliquoring of goat skin wet blue was conducted according to a typical leather processing procedure, in which the dosages of chemicals were calculated by the weight of wet blue skin. A piece of goat skin wet blue about 10 cm × 10 cm was weighted and put into a drum. The wet blue skin was degreased using 200% water and 0.2% commercial degreasant at 30 °C for 60 min. Then, the degreased wet blue skin was washed twice by 200% water at 30 °C for 15 min. Into the drum, 200% fresh water was added, and the pH of the liquid was adjusted to 3.4 using 0.4% formic acid for 10 min. Then, 4% chrome powder was added at 30 °C for re-chroming 60 min later, it was basified by using 0.2% sodium formate to pH 4.0 for 30 min. After being washed, the re-chromed leather was neutralized fully on its cross-section in 200% water by 0.4% HCOONa and 0.1% NaHCO_3_ to pH 6.5 at 40 °C for 30 min. The water was drained off, and the leather was washed again as per the method above. Subsequently, the leather was fat-liquored in 200% water containing 10% AFSCO for 90 min at 50 °C. At the end of fatliquoring, the oil was fixed by decreasing the pH of liquor to 4.0 using 1.0% formic acid.

#### 2.4.2. Tracing of Fatliquoring Agent

After fatliquoring, the leather was sampled and cut vertically to obtain a slice with a thickness of 25 μm using a freezing microtome (CM1950, Leica, Wetzlar, Germany). The slice was loaded on a glass slide and immersed in a diluted alkali solution at pH 10 for 30 s. Then, the section of the sample was observed using an inverted fluorescence microscope (IX73, Olympus, Tokyo, Japan) to locate AFSCO in leather. Meanwhile, the Image J software was used to analyze the fluorescence micrograph, in order to censure the relative contents of AFSCO in the cross section of leather.

Additionally, leather slices were stained with Sudan IV or Nile Blue sulphate, and their processes were as follows. In a mixed solution of 70% (*v*/*v*) ethanol solution and acetone, 0.5 g of Sudan IV was dissolved. The slice was immersed in the above-mentioned Sudan IV solution to be stained for 1 min, and then washed successively with 50% (*v*/*v*) ethanol solution, 30% (*v*/*v*) ethanol solution and deionized water. On the other hand, the slice was immersed in 10 g/L of Nile Blue sulphate solution at 60 °C for 10 min, and then washed twice with deionized water, which was followed by differentiating for 30 min in a 1% (*w*/*w*) acetic acid solution. Finally, the photomicrographs of the stained slices were taken using a digital microsystem (DVM6 A, Leica, Heerbrugg, Switzerland).

#### 2.4.3. Preparation of Samples for Mass Transfer Investigations

The mass transfer of AFSCO at different times was investigated by using the fluorescent tracing method established above. The leather was sampled at 30, 60 and 90 min, respectively, in which the dosage of fatliquoring agent was 15% and other procedures were the same as in Section 2.4.1. Furthermore, in order to research the distribution of fatliquoring agent with different dosages, 5%, 10% and 20% AFSCO were used in the fatliquoring of leather, respectively. After fatliquoring, the leather was sliced and analyzed as mentioned above.

## 3. Results and Discussion

### 3.1. FTIR Spectra

The FTIR spectra of castor oil, CCO and AFCO are shown in Figure 2. As shown in Figure 2a, the adsorbent band around 3400 cm^−1^ was attributed to the stretching vibrations of hydroxyl groups on the long fat chains of castor oil, while the peaks at 2928 cm^−1^ and 2855 cm^−1^ corresponded to the asymmetric stretching vibrations of CH_3_ and CH_2_ on castor oil [18]. The characteristic peak at 1745 cm^−1^ was due to the stretching vibrations of C=O on ricinoleic acid [17]. After modification, a carboxylate castor oil was obtained and its FTIR spectrum is shown in Figure 2b. The characteristic band of the hydroxyl group at 3400 cm^−1^ almost disappeared and a new peak corresponding to C=C at 1641 cm^−1^ appeared, which proved the esterification of castor oil with maleic anhydride. When the carboxylate castor oil was labeled with 5-aminofluorescein, a new peak was observed at 1590 cm^−1^ (shown in Figure 2c), a characteristic peak of the amide II band [19] that demonstrated the formation of amide arising from the reaction of the carboxyl group with the amine group within 5-aminofluorescein. The FTIR analysis verified that the fluorescent molecule was grafted successfully onto castor oil.

### 3.2. ^1^H NMR Analysis

The chemical characteristics of the castor oil, CCO as well as AFCO were further measured by ^1^H NMR. As shown in Figure 3A, the peaks at δ 0.69–0.85 ppm were assigned to the protons of the terminal methyl groups of castor oil, while the chemical shift at 1.04–1.67 ppm was attributed to the protons of methylene groups. The chemical shifts between 1.91 and 2.2 ppm showed the protons of the –CH_2_C=O groups. The peak at 3.51 ppm was not observed on the ^1^H NMR spectra of natural oils without the hydroxy group, such as rapeseed oil [20], *Afzelia africana* aril cap oil [3] and *Sesamum indicum* L. seed oil [21]; thus, it might be the proton of –OH of castor oil. The protons of methylene groups on the glyceride moiety gave δ at 4.01–4.20 ppm, while peaks at 5.15–5.47 ppm were assigned to the protons of CH=CH on the fatty acid chains of castor oil. After being modified with maleic anhydride, shown in Figure 3B, new peaks at δ 6.2–6.4 ppm and 4.9 ppm were attributed to the CH=CH and O=C–OH of maleate, respectively. The ^1^H NMR spectrum of AF-labeled castor oil is shown in Figure 3C. The protons on CH=CH of the benzene skeleton were observed at δ 6.81 ppm. Moreover, the single peak at 3.51 ppm became multiple peaks due to the introduction of the phenolic hydroxyl group of 5-aminofluorescein. The results of ^1^H NMR further confirmed that the castor oil was successfully labeled by a fluorescent molecule.

### 3.3. DSC Investigation

Generally, the chemical composition of the oils influences their thermal properties, such as the glass transition temperature, crystallization temperature and so on [22]. Therefore, the thermal properties of castor oil, CCO and AFCO were detected using DSC, and their DSC curves are shown in Figure 4a–c. Castor oil is a special vegetable oil with good low-temperature fluidity and no crystallization temperature was recorded on its DSC curve which was different from other natural oils such as sunflower oil [23]. Nevertheless, the pour point temperature of castor oil was observed at −53.12 °C, as shown in Figure 4a, which was very close to the previous report (−55.06 °C) recorded by L.A. Quinchia [24]. After reacting with maleic anhydride, the pour point temperature of carboxylate castor oil was decreased to −63.59 °C. This was because the molecular mobility as well as the unsaturation of the product increased with the grafting of maleic acid. However, the pour point temperature of AFCO increased to −47.63 °C, which might be due to the limitation of aromatic aminofluorescein on the rotation freedom of the oil molecule. This was in agreement with the instance that the pour point temperature of the mineral oil shifted to a high temperature under the effect of aromatic composition [23,25]. The change in the pour point temperatures of castor oils before and after modification was ascribed to the transformation of their molecular structures, which verified the successful synthesis of AF-labeled castor oil.

### 3.4. Fluorescence Emission Spectra

In order to assess the applicable conditions of AFCO for establishing the fluorescent tracing technique of the fatliquoring agent in leather, the effect of pH on the fluorescence intensity of AFCO was researched and the results are shown in Figure 5. It could be observed that the fluorescence intensity of AFCO increased with the increase in pH value. The intensity of the emission maximum was very low at pH 6, while it increased to 70.74 and 159.85 a.u. when the pH value increased to 7 and 8, respectively. Furthermore, the intensity of the emission maximum was increased greatly to 443.87 and 599.01 a.u. at pH 9 and 10, respectively. The optical images of AF-labeled castor oil at different pH are shown in the insets of Figure 5, and obvious green fluorescence was observed at pH levels 8, 9, and 10 when the solutions were irradiated by UV light. This proved that AFCO was pH-sensitive, which was similar to the AF-labeled acrylic resin re-tanning agent prepared by Zeng [14]. Therefore, the leather sample would be dealt with in dilute sodium hydroxide at pH 10 to receive intense fluorescence signals.

### 3.5. Observation of the Distribution of Fatliquoring Agent in Leather

The distribution of fatliquoring agents in leather was investigated by the Sudan IV staining method, Nile Blue sulphate staining method and fluorescent tracing technology. The results are displayed in Figure 6a–c, respectively. Generally, the oil-soluble dye Sudan IV would combine with the fatty acid of a fatliquoring agent to reveal the distribution of fatliquoring agent in leather. However, as shown in Figure 6a, it did not show the trace of fatliquoring agent clearly. Compared with Sudan IV, using the Nile Blue sulphate staining method, the distribution of the fatliquoring agent in leather was very clear. It was found that the fatliquoring agent was distributed mainly in the flesh layer of the leather. However, some oval, stained areas were found, which were distinguished from their surrounding areas. These stained areas were natural adipose glands of goat skin, rather than the regions full of additive fatliquoring agents. In other words, the staining method could not distinguish the innate fats located in skin from extraneous fatliquoring agent. According to the change in fluorescent intensity shown in Figure 6c, it was also revealed that the fatliquoring agent was mainly distributed in the flesh layer and rarely in the grain layer of the leather when its dosage was relatively insufficient (10%). Meanwhile, no adipose glands were observed on the fluorescent tracing image, which proved the fluorescent tracing technology could overcome the shortcomings of the staining methods above. It could be concluded that the fluorescent tracing method for fatliquoring agents based on AF-labeled castor oil was successfully established.

### 3.6. Mass Transfer of Fatliquoring Agent in Leather

Relying on the fluorescent tracing technology for the fatliquoring agents constructed above, the mass transfers of AFSCO at different times were studied in detail. The results are shown in Figure 7a–d. When the AFSCO was added to the drum for 30 min, the fluorescent signal was distributed primarily in the flesh layer of the leather, while only a thin band with a fluorescent signal on the grain layer was observed. It was found that the fatliquoring agent penetrated the leather mainly from the flesh side. This was because of the woven structure of the collagen fibers on the flesh side, which was much looser than on the grain side, and the stronger capillary action occurred on the flesh side. It was further quantified using Image J, and the result is shown in Figure 7d. It can be observed that the penetrating distance of AFSCO from the flesh side was about 0.35 mm in 30 min, but its penetrating distance from grain side was around 0.15 mm. The relative content of AFSCO on the flesh side was almost higher than 60% compared to the grain side, which was less than 60% (shown as the blue water-full curve in Figure 7d). With the increase in the fatliquoring time, the contents of AFSCO both on the flesh side and the grain side were increased. Especially in the central part of leather sample, the areas with high fluorescent intensities increased. After 90 min, fluorescent signals dispersed on the whole vertical section of the leather, which meant the fatliquoring agent had distributed throughout the leather. However, the dispersion of fatliquoring agent in leather was not uniform. Remarkably, the content of the fatliquoring agent on the flesh side was much more than on the grain side.

It should be noted that the content of AFSCO in the location where it was 0.2–0.3 mm to the grain side was the lowest and had little change in 90 min. From this special region extending outward to both the grain side and the flesh side, all the contents of AFSCO increased gradually. It can be concluded that the dosage (15%) of fatliquoring agent in this experiment was insufficient for its even distribution on the whole section of the leather. It can also be inferred that the mass transfer rate of the fatliquoring agent penetrating from the flesh side was 4–5 times higher compared to the grain side.

Aside from the effect of time, the dosage of fatliquoring agent was an important factor that greatly influenced the quality of leather. Thus, the distribution of fatliquoring agents with different dosages was investigated further. The fluorescence images and quantification analysis are shown in Figure 8. When the dosage of AFSCO was 5%, only about one-quarter of the region in the cross-section area of the crust leather had adsorbed the fatliquoring agent. While the fatliquoring agent was mainly concentrated in the flesh layer, the inner part of the crust leather had hardly any fatliquoring agent. As the dosage of AFSCO was increased to 10%, the distribution of fatliquoring agent in leather was greatly modified. When the dosage of fatliquoring agent was 20%, its distribution in leather became uniform, as shown in Figure 8c, and the green water-full curve in Figure 8d.

## 4. Conclusions

Firstly, an AF-labeled castor oil was synthesized by the reaction of castor oil sequentially with maleic anhydride and 5-aminofluorescein. The FTIR spectra, ^1^H NMR spectra and DSC curves of the starting material and product proved that the fluorescent molecule had been grafted successfully onto castor oil. The AF-labeled castor oil was pH-sensitive and a strong fluorescent signal would be obtained at pH 10. Based on the AF-labeled castor oil, a fluorescent tracing technique, for visually investigating and quantifying the mass transfer of fatliquoring agent in leather, was established. Compared with Sudan IV and Nile Blue sulphate staining methods, the penetration and distribution of fatliquoring agents in leather could be analyzed preferably by using the fluorescent tracing technique. This technique can be used as a real-time monitor for investigating and quantifying the mass transfer of fatliquoring agent, which is beneficial for lucubrating the fatliquoring mechanism and its structure–activity relationship of fatliquor.

## Figures and Tables

**Figure 1 materials-15-01167-f001:**
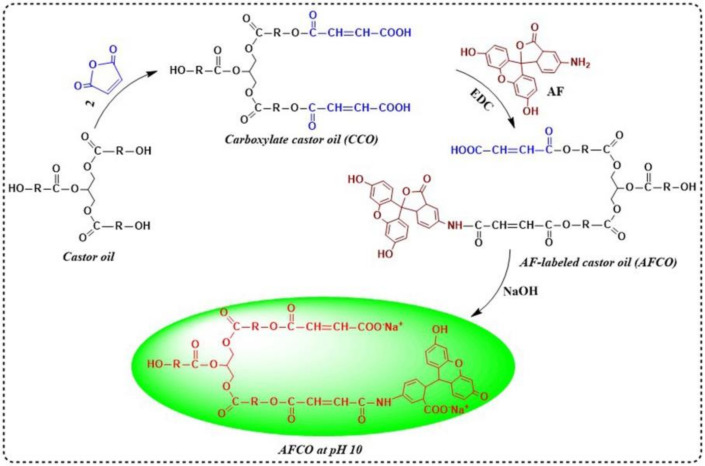
Synthesis procedures of fluorescently labeled tracer (AFCO).

**Figure 2 materials-15-01167-f002:**
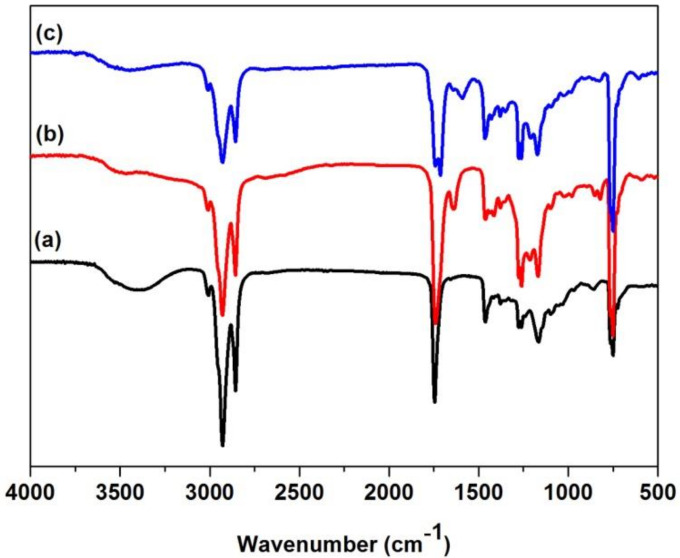
The FTIR spectra of samples: (**a**) castor oil, (**b**) CCO and (**c**) AFCO.

**Figure 3 materials-15-01167-f003:**
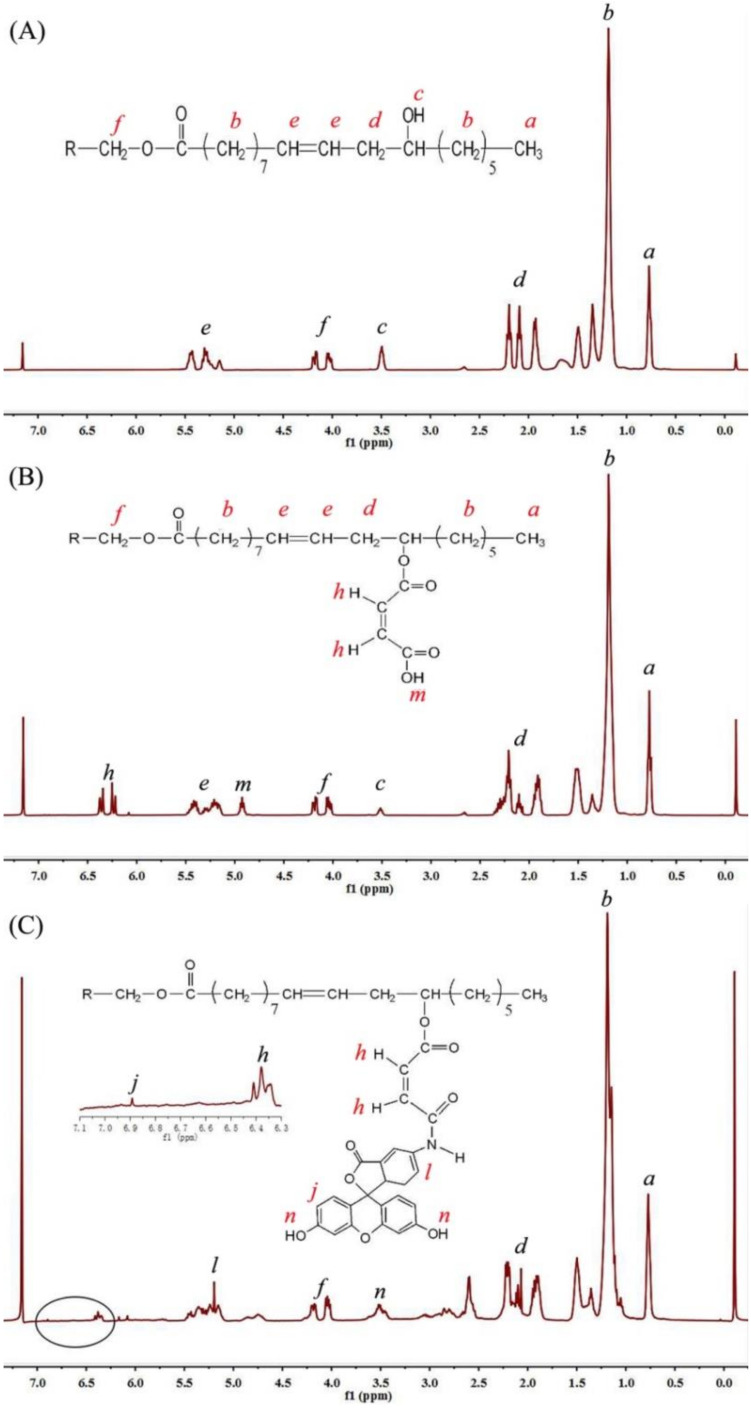
^1^H NMR of: (**A**) castor oil, (**B**) CCO and (**C**) AFCO.

**Figure 4 materials-15-01167-f004:**
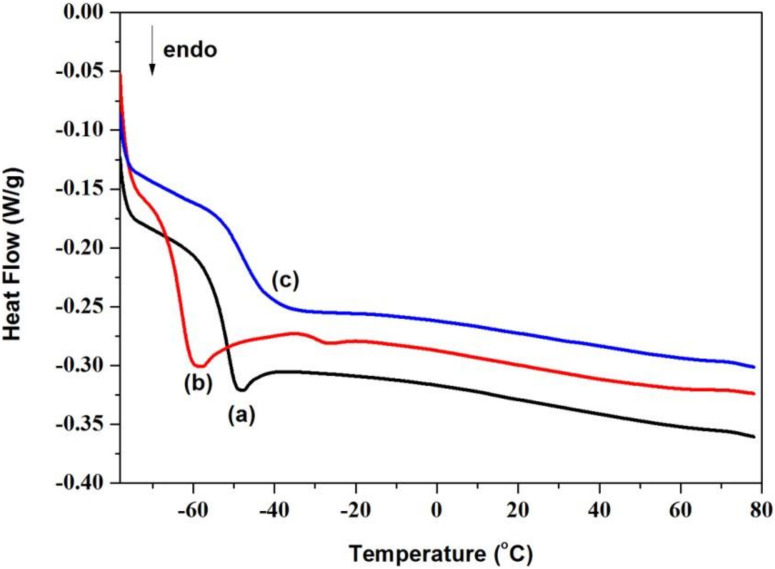
DSC curves of oils: (**a**) castor oil, (**b**) CCO and (**c**) AFCO.

**Figure 5 materials-15-01167-f005:**
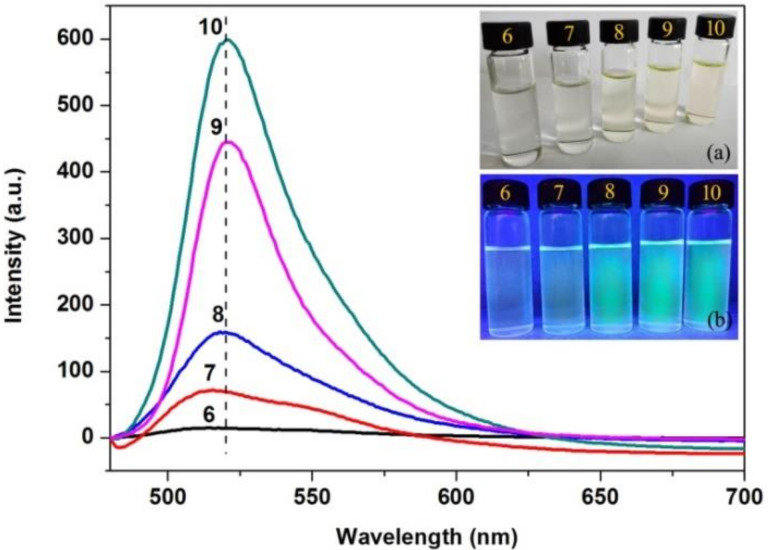
Fluorescence emission spectra of AFCO at different pH, and the insets are the optical images of AFCO solutions at different pH: (**a**) under visible light and (**b**) under ultraviolet light.

**Figure 6 materials-15-01167-f006:**
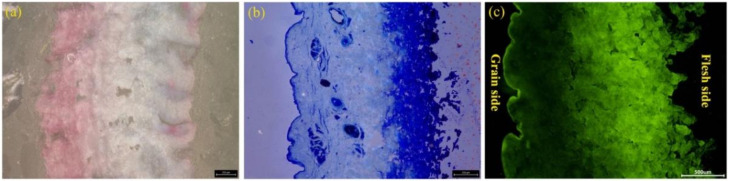
Images of slices with different tracing method for fatliquoring agent distributed in leather: (**a**) Sudan IV stained, (**b**) Nile Blue sulphate stained and (**c**) fluorescent tracing technology by AFCO.

**Figure 7 materials-15-01167-f007:**
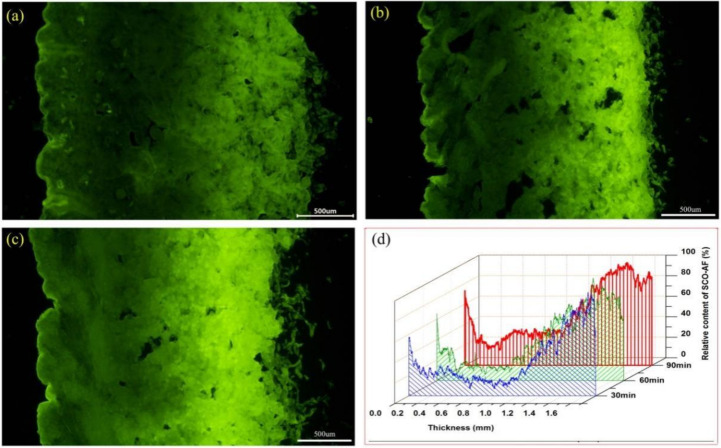
Fluorescence micrographs of AFSCO transferred into leather at different time: (**a**) 30 min, (**b**) 60 min, (**c**) 90 min and (**d**) quantification analysis of fluorescence micrographs.

**Figure 8 materials-15-01167-f008:**
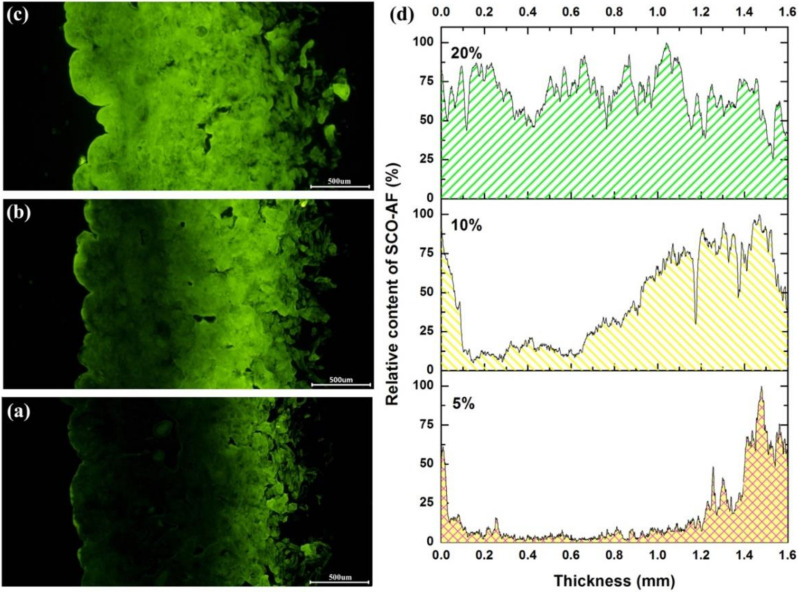
Fluorescence micrographs of different amount of AFSCO distributing in leather: (**a**) 5%, (**b**) 10%, (**c**) 20% and (**d**) quantification analysis of fluorescence micrographs.

## Data Availability

Not applicable.

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
