# Peer review of "A Fluorescent Tracer Based on Castor Oil for Monitoring the Mass Transfer of Fatliquoring Agent in Leather"

_materials, 2022, doi:10.3390/ma15031167_

Round 1

Reviewer 1 Report

The manuscript is very important for the understanding and for sustainable and efficient new technologies design for leather industry. The works add an advance in the area of fluorescent labeling of chemical materials. Some minor corrections are proposed below.

Abstract

fibers are split effectively/ fatliquoring process envelop the splitted collagen fibers and fill the voids. If the collagen fibers are not splitted by tanning process, fatliquoring can’t do much in the splitting. In introduction you say this “After being fat-liquored, the collagen fibers are covered by very thin oil membranes”.

Introduction

fat-liquoring/ I recommend to use fatliquoring, the most known term

It is a method of checking the penetration of chemical materials by making longitudinal sections of leather surface (1-2 slices) and analyzing by chemical methods of chromium, fatliquors content. The concentration of fatliquor/chromium in leather thickness can be evaluated. This classical method is worth to be mentioned. SEM/EDX  elemental composition mapping can be also used as a semi quantitative method.

Materials

Pg.2/97 EDC/ to explain

Pg.3/98 Goat wet blue/goat skins in wet-blue state

Pg.3/99 chrome powder/basic chromium sulphate You should present the characteristics as well as for the fatliquors and to mention they are technical grade. For all chemicals it should be presented the concentrations.

Fig 1/ castor oil or sulfated castor oil ?

2.2.1. Preparation of the fluorescent tracer/ The treatment with NaOH is not described

2.2.2. Preparation of fluorescent fatliquoring agent/ the missing of time of interaction of AFCO with sulfated castor oil

Pg.4/129 dried castor oil/ or dried sulfated castor oil ?

2.4.1. The fatliquoring process of goat wet-blue/2.4.1. The fatliquoring process of goat skin wet-blue

Pg.4/150 goat wet-blue/ goat skin wet-blue

Pg.4/152/3 the wet-blue/the wet-blue skins

Pg.4/154 degreased wet blue/ degreased wet blue skin

Pg.4/157 retanning/ rechromed

Pg.4/158 the retanned leather/ the rechromed leather

Pg.4/159 HCOONa and 0.1% NaHCO3 to pH 6.5 at 40 oC for 30min/you have to mention the cross section neutralization

2.4.3. Mass transfer of fatliquoring agent/ Maybe the title has to be Preparation of samples for mass transfer investigations

3.1. FTIR spectra

Pg 5/191 castor oil/sulfated castor oil

You have to mention inside the text Fig.2c

3.2. 1H NMR analysis

You have to mention the Fig 3B inside the text

Author Response

Dear editors and reviewers:

Thanks for your comments concerning our manuscript numbered “materials-1562935”. Those comments are all valuable and very helpful for improving our paper. We have studied comments carefully and have made correction which we hope meet with approval.

(1). fibers are split effectively/ fatliquoring process envelop the splitted collagen fibers and fill the voids. If the collagen fibers are not splitted by tanning process, fatliquoring can’t do much in the splitting. In introduction you say this “After being fat-liquored, the collagen fibers are covered by very thin oil membranes”.

Response: Yes, as you mentioned, if the collagen fibers are not splitted by tanning process, fatliquoring can’t do much in the splitting. Therefore, the fatliquoring process is generally after retanning process. And the expression has been modified as “After being fat-liquored, the splitted collagen fibers of retanned leather are enveloped further by very thin oil membranes”.

(2). fat-liquoring/ I recommend to use fatliquoring, the most known term

Response: OK. Every word of ”fat-liquoring” in this manuscript has been revised as “fatliquoring”.

(3). It is a method of checking the penetration of chemical materials by making longitudinal sections of leather surface (1-2 slices) and analyzing by chemical methods of chromium, fatliquors content. The concentration of fatliquor/chromium in leather thickness can be evaluated. This classical method is worth to be mentioned. SEM/EDX elemental composition mapping can be also used as a semi quantitative method.

Response: It is a good advice, and we have revised out manuscript. “Scanning Electron Microscope-Energy Dispersive X-ray (SEM-EDX) analysis is a semi quantitative method in mapping metal elements composition such as chrome and aluminum in leather samples [7]. Unfortunately, this method is not suitable for the analysis of organic fatliquoring agent, mainly composed of carbon, hydrogen and oxygen.” A new reference was cited and the serial numbers of other references have been modified.

(4). Pg.2/97 EDC/ to explain

Response: EDC is the abbreviation of 1-(3-Dimethylaminopropyl)-3-ethylcarbodiimide hydrochloride, and its full form has been added in our manuscript.

(5). Pg.3/98 Goat wet blue/goat skins in wet-blue state

Response: This phrase of “Goat wet blue” has been changed as “Goat skin in wet-blue state”.

(6). Pg.3/99 chrome powder/basic chromium sulphate You should present the characteristics as well as for the fatliquors and to mention they are technical grade. For all chemicals it should be presented the concentrations.

Response: Yes, it is important. The expression of “The commercial chrome powder and degreasing agent” had been instead by “A technical grade basic chromium sulphate (CTS, basicity, 33%; Cr2O3 content, 25%) were supplied from Brother Enterprise Holding Co., Ltd. (Zhejiang, China), and a technical grade nonionic degreasing agent (FB, 40%) was offered by Leahou Light Industrial of New Material Co., Ltd. (Shandong, China)”.

(7). Fig 1/ castor oil or sulfated castor oil?

Response: The raw material used in the preparation of fluorescent tracer illustrated in Figure 1 was castor oil, it was not sulfated castor oil.

(8). 2.2.1. Preparation of the fluorescent tracer/ The treatment with NaOH is not described

Response: NaOH was really not used in the preparation and purification processces of the fluorescent tracer (AFCO shown in Figure 1). The pure AFCO is a carboxylic acid and its fluorescent property at acidic condition is very weak. And it has to be used in alkaline condition for receiving intensive fluorescence signal. The final step in Figure 1 was used to show the application condition for fluorescent tracer.  

(9). 2.2.2. Preparation of fluorescent fatliquoring agent/ the missing of time of interaction of AFCO with sulfated castor oil

Response: It is a good advice and we supplied relevant details. “100 mg of AFCO dissolved in 0.5 mL butoxyethanol was mixed with 20 g sulfated castor oil on Vortex-Genie 2 (Scientific Industries, USA) for 1 min to prepare a fluorescent fatliquoring agent”.

(10). Pg.4/129 dried castor oil/ or dried sulfated castor oil?

Response: Before FTIR analysis, the castor oil, CCO and AFCO should be dried in oven at 110oC for 4h to remove potential trace amounts of water. Thus it was castor oil in this section.

(11). 2.4.1. The fatliquoring process of goat wet-blue/2.4.1. The fatliquoring process of goat skin wet-blue.

Response: It has been revised.

(12). Pg.4/150 goat wet-blue/ goat skin wet-blue

Response: It has been revised.

(13). Pg.4/152/3 the wet-blue/the wet-blue skins

Response: It has been revised.

(14). Pg.4/154 degreased wet blue/ degreased wet blue skin

Response: It has been revised.

(15). Pg.4/157 retanning/ rechromed

Response: “retanning“ was replaced by “re-chroming”.

(16) Pg.4/158 the retanned leather/ the re-chromed leather

Response: It has been revised.

(17). Pg.4/159 HCOONa and 0.1% NaHCO3 to pH 6.5 at 40 oC for 30min/you have to mention the cross section neutralization

Response: We have modified it as: After being washed, the re-chromed leather was neutralized fully on its cross section in 200% water by 0.4% HCOONa and 0.1% NaHCO3 to pH 6.5 at 40 oC for 30min.

(18). 2.4.3. Mass transfer of fatliquoring agent/ Maybe the title has to be Preparation of samples for mass transfer investigations.

Response: The title of 2.4.3 has been changed according to this good suggestion.

(19). 3.1. FTIR spectra

Pg 5/191 castor oil/sulfated castor oil

Response: Here it is castor oil rather than sulfated castor oil.

(20). You have to mention inside the text Fig. 2c

Response: Related information has been given in the sentence of “a new peak was observed at 1590 cm-1 (shown in Fig. 2c) and it was a characteristic peak of amide II band”.

(21). 3.2. 1H NMR analysis

You have to mention the Fig 3B inside the text

Response: It has been modified and the third sentence in page 6 was revised as: After being modified with maleic anhydride, shown as Figure 3B, new peaks at δ 6.2-6.4 ppm and 4.9 ppm were attributed to CH=CH and O=C-OH of maleate, respectively.

Reviewer 2 Report

Abstract

  • Page 1 Line no 9, “Fatliquoring is one of the most important processes in leather making, in which the collagen fibers are split effectively while the crust acquired good softness and mechanical strength.” may be written as “Fatliquoring is one of the most important processes in leather making, in which the collagen fibers are splitted effectively while the crust acquired good softness and mechanical strength.”
  • Page 1 Line no 16, “The crucial fluorescent tracer was synthesized successfully by the reaction of castor oil successively with maleic anhydride and 5-aminofluorescein, which was confirmed by FTIR, 1H NMR and DSC.” may be written as “The crucial fluorescent tracer was synthesized favorably by the reaction of castor oil successively with maleic anhydride and 5-aminofluorescein, which was confirmed by FTIR, 1H NMR and DSC.”
  • Page 1 Line no 19, “And then it was applied to monitor the fatliquoring process real-timely.” may be written as “And then it was applied to monitor the fatliquoring process real-time.”

Introduction

  • Page 1 Line no 29, “Without the lubrication of the natural fats, the tanned collagen fibers would stick together when the water was lost and the wet blue is also easy to be cracking.” may be written as “Without the lubrication of the natural fats, the tanned collagen fibers will stick together when the water will lose and cracking of wet blue takes place easily.”
  • Page 1 Line no 36, “It should be noted that the improvement of fat-liquoring agent to the mechanical properties of leather is strongly associated with their distribution in the crust.” may be written as “It should be noted that the improvement of fat-liquoring agent in the mechanical properties of leather is strongly associated with its distribution in the crust.”
  • Page 1 Line no 41, “Additionally, the chemicals combined previously with collagen fibers would influence the penetration and combination of subsequent chemicals, and a typical example is the fading effect to dye of acrylic polymers.” may be written as “Additionally, the chemicals combined previously with collagen fibers will influence the penetration and combination of subsequent chemicals, and a typical example is the fading effect to dye of acrylic polymers.”
  • Page 1 Line no 43, “It is undoubtedly that the penetrating and distributing of fat-liquoring agent in the whole cross section of leather is a very complex process.” may be written as “It is undoubtedly that the penetration and distribution of fat-liquoring agent in the whole cross section of leather is a very complex process.”
  • Page 2 Line no 46, “Therefore the investigation of the mass transfer processes of fat-liquoring agent in wet blue as well as its distribution is crucial to understanding its structure-activity relationship, which is very important for designing and developing of new fat-liquoring agent.” may be written as “Therefore, the investigation of the mass transfer processes of fat-liquoring agent in wet blue as well as its distribution are crucial to understand its structure-activity relationship, which is very important for designing and developing of new fat-liquoring agent.”
  • Page 2 Line no 49, “However, the researches in this field were mainly focus on developing new kinds of fatliquoring agents and rarely on their mass transfer and distribution in wet blue.” may be written as “However, the researches in this field are mainly focused on developing new kinds of fatliquoring agents and rarely on their mass transfer and distribution in wet blue.”
  • Page 2 Line no 53, “But these staining methods are relatively complicated in sample preparing and the slice is easy to be broken in dying process. What more important is that these methods are difficult in monitoring the mass transfer of fat-liquoring agents in real time.” may be written as “But these staining methods are relatively complicated in sample preparation and the slice is easy to be broken in dying process. What more important is that these methods are difficult in monitoring the mass transfer of fat-liquoring agents in real-time.”
  • Page 2 Line no 74, “However, few papers about investigating the mass transfer of fatliquoring agent in leather by using fluorescent tracing method were reported.” may be written as “However, few papers about investigating the mass transfer of fatliquoring agent in leather by using fluorescent tracing method, were reported.”
  • Page 2 Line no 81, “While what important for researching its mass transfer is to develop a fluorescent tracer which has similar structure with sulfated castor oil.” may be written as “While, what important for researching its mass transfer is to develop a fluorescent tracer which has similar structure with sulfated castor oil.”
  • Page 2 Line no 90, “Finally, the mass transfer of sulfated castor oil containing 0.5% fluorescent tracer in goat wet blue was researched in details.” may be written as “Finally, the mass transfer of sulfated castor oil containing 0.5% fluorescent tracer in goat wet blue was investigated in details.”

Materials and Methods

  • Page 3 Line no 109, “Then 19.6 grams of maleic anhydride was introduced into the system at 110 oC and reacted for 4 h to obtained carboxylate castor oil (CCO).” may be written as “Then 19.6 grams of maleic anhydride was introduced into the system at 110 oC and reacted for 4 to obtain carboxylate castor oil (CCO).”
  • Page 3 Line no 112, “After the carboxylate castor oil was dissolved completely, 34.7 mg AF was used to react with the modified oil above for 114 min under the catalyzing of 19.1 mg EDC.” may be written as “After the carboxylate castor oil was dissolved completely, 34.7 mg AF was used to react with the above modified oil for 60 min under the catalyzation of 19.1 mg EDC.”
  • Starting tab spacing of headings 2.1, 2.2.1 and 2.2.2 is not equal.
  • Page 3 Line no 127, Characterizations should be characterization.
  • Page 4 Line no 129, “The dried castor oil, CCO and AFCO was dropped on a BaF2 wafer for FTIR spectra analysis, respectively.” May be written as “The dried castor oil, CCO and AFCO was dropped on a BaF2 wafer for FTIR spectra analysis, respectively.”
  • Page no 4 Line no 134, “About 15 mg of castor oil, CCO and AFCO was respectively dissolved in 0.5 mL of CDCl3 solvent in a 5 mm-diameter sample tube for NMR of 1H (AVANCE II 400, Bruker, Germany) analysis.” may be written as “About 15 mg of castor oil, CCO and AFCO respectively were dissolved in 0.5 mL of CDCl3 solvent in a 5 mm-diameter sample tube for NMR of 1H (AVANCE II 400, Bruker, Germany) analysis.”
  • Page 4 Line no 138, “About 10 mg of castor oil, CCO and AFCO was respectively sealed in an aluminum pan with an empty sealed aluminum pan as the reference.” may be written as “About 10 mg of castor oil, CCO and AFCO respectively were sealed in an aluminum pan with an empty sealed aluminum pan as the reference. “
  • Page 4 Line no 139, “The DSC curves of oil sample above were conducted on DSC250 (TA, USA) in the temperature ranged from -80 °C to 80°C with a constant heating rate of 5 °C/min under a nitrogen flow.” may be written as “The DSC curves of above oil sample were conducted on DSC250 (TA, USA) in the temperature ranged from -80 °C to 80°C with a constant heating rate of 5 °C/min under a nitrogen flow.”
  • Headings from 2.3.1 to 2.3.4 either capitalized each first word or not.
  • Page 4 Line no 155, “200% fresh water was added into the drum and the pH of the liquid was adjusted to 3.4 by using 0.4% formic acid in 10 min.” may be written as “200% fresh water was added into the drum and the pH of the liquid was adjusted to 3.4 by using 0.4% formic acid for 10 min.”
  • Page 4 Line no 156, “And then, 4% chrome powder was added at 30oC for retanning 60 min later, it was basified by using 0.2% sodium formate to pH 4.0 in 30 min.” may be written as “And then, 4% chrome powder was added at 30oC for retanning 60 min later, it was basified by using 0.2% sodium formate to pH 4.0 for 30 min.”
  • Page 4 Line no 160, “Subsequently, the leather was fat-liquored in 200% water containing with 10% AFSCO for 90 min at 161o” may be written as “Subsequently, the leather was fat-liquored in 200% water containing with 10% AFSCO for 90 min at 161oC.”
  • Page 4 Line no 164, “At the end of fatliquoring, the oil was fixed through decreasing the pH of liquor to 4.0 by 1.0% formic acid.” may be written as “At the end of fatliquoring, the oil was fixed by decreasing the pH of liquor to 4.0 by 1.0% formic acid.”
  • Page 4 Line no 174, “The slice was immersed into the Sudan IV solution above to be stained for 1min, and then washed successively by 50% (v/v) ethanol solution, 30% (v/v) ethanol solution and deionized water.” may be written as “The slice was immersed into the above-mentioned Sudan IV solution to be stained for 1min, and then washed successively by 50% (v/v) ethanol solution, 30% (v/v) ethanol solution and deionized water.”
  • Page 5 Line no 183, “The leather was sampled respectively at 30, 60 184 and 90 min, in which the dosage of fatliquoring agent was 15% and other procedures were the same as section 2.4.1.” may be written as “The leather was sampled respectively at 30, 60 184 and 90 min, in which the dosage of fatliquoring agent was 15% and other procedures were the same as in section 2.4.1.”
  • Page 5 Line no 185, “Furthermore, in order to research the distribution of fatliquoring agent with different dosage, 5%, 10% and 20% AFSCO was respectively used in the fatliquroing of leather.” may be written as “Furthermore, in order to research the distribution of fatliquoring agent with different dosages, 5%, 10% and 20% AFSCO were respectively used in the fatliquroing of leather.”

Results and Discussion

  • Page 5 Line no 197, “The characteristic band of hydroxyl group at 3400 cm-1 almost disappeared and a new peak corresponding to C=C at 1641 cm-1 appeared, which proved the esterification of castor oil with maleic anhydride reacted.” may be written as “The characteristic band of hydroxyl group at 3400 cm-1 almost disappeared and a new peak corresponding to C=C at 1641 cm-1 appeared, which proved the esterification of castor oil with maleic anhydride reacted.”
  • Page 5 Line no 209, “As shown in Fig. 3A, the peaks at δ 0.69-0.85 ppm are assigned to the protons of terminal methyl groups of castor oil, while the chemical shift at 1.04-1.67 ppm attributed to the protons of methylene groups.” may be written as “As shown in Fig. 3A, the peaks at δ 0.69-0.85 ppm were assigned to the protons of terminal methyl groups of castor oil, while the chemical shift at 1.04-1.67 ppm was attributed to the protons of methylene groups.”
  • Page 5 Line no 211, “The chemical shift between 1.91 and 2.2 ppm showed the protons of -CH2C=O groups.” may be written as “The chemical shifts between 1.91 and 2.2 ppm showed the protons of -CH2C=O groups.”
  • Page 6 Line no 230, “Castor oil was a special vegetable oil with good low-temperature fluidity, and no crystallization temperature was recorded on its DSC curve, which different from other natural oil such as sunflower oil.” may be written as “Castor oil was a special vegetable oil with good low-temperature fluidity, and no crystallization temperature was recorded on its DSC curve, DSC curve, which was differed from other natural oil such as sunflower oil.”
  • Page 7 Line no 235, “After reacting with maleic anhydride, the pour point temperature of carboxylate castor oil decreased to -63.59 o” may be written as “After reacting with maleic anhydride, the pour point temperature of carboxylate castor oil was decreased to -63.59 236 oC.”
  • Page 7 Line no 236, “It was because the molecular mobility as well as the unsaturation of the product increased as the grafting of maleic acid.” may be written as “It was because the molecular mobility as well as the unsaturation of the product was increased with the grafting of maleic acid.”
  • Page 7 Line no 237, “However, the pour point temperature of AFCO increased to -47.63 oC, which might be due to the limitation of aromatic aminofluorescein on the rotation freedom of oil molecule.” may be written as “However, the pour point temperature of AFCO was increased to -47.63 oC, which might be due to the limitation of aromatic aminofluorescein on the rotation freedom of oil molecule.”
  • Page 7 Line no 250, “It could be observed that the fluorescence intensity of AFCO increased with the increase of pH value.” may be written as “It could be observed that the fluorescence intensity of AFCO was increased with the increase of pH value.”
  • Page 7 Line no 252, “The intensity of the emission maximum was very low at pH 6, while it increased to 70.74 and 159.85a.u.when the pH value increased to 7 and 8, respectively.” may be written as “The intensity of the emission maximum was very low at pH 6, while it was increased to 70.74 and 159. 85a.u. (spacing) when the pH value was increased to 7 and 8, respectively.”
  • Page 7 Line no 254, “Furthermore, the 254 intensity of the emission maximum increased greatly to 443.87 and 599.01a.u.at pH 9 and 10, respectively.” may be written as “Furthermore, the 254 intensity of the emission maximum was increased greatly to 443.87 and 599.01a.u. (spacing) at pH 9 and 10, respectively.”
  • Page 8 Line no 264, “3.5. Observation the distribution of fatliquoring agent in leather” may be written as “3.5. Observation of distribution of fatliquoring agent in leather”
  • Page 8 Line no 277, “According to the change of fluorescent intensity, shown as Fig. 6c, it also revealed that the fatliquoring agent mainly distributed in the flesh layer and rarely in grain layer of the leather when its dosage was relatively insufficient (10%).” may be written as “According to the change of fluorescent intensity, shown as Fig. 6c, it was also revealed that the fatliquoring agent mainly distributed in the flesh layer and rarely in grain layer of the leather when its dosage was relatively insufficient (10%).”
  • Page 8 Line no 289, “Relying on the fluorescent tracing technology for fatliquoring agent constructed above, the mass transfers of AFSCO at different time were studied in details, which results were shown in Fig. 7a-d.” may be written as “Relying on the fluorescent tracing technology for fatliquoring agent constructed above, the mass transfers of AFSCO at different time were studied in details, whose results were shown in Fig. 7a-d.”
  • Page 9 Line no 293, “It reflected that the fatliquoring agent penetrated into the leather mainly from flesh side, which was because that the woven structure of collagen fibers on flesh side was much looser than it on grain side and stronger capillary action occurred on the flesh side.” may be written as “It was reflected that the fatliquoring agent penetrated into the leather mainly from flesh side, which was because of woven structure of collagen fibers on flesh side which was much looser than it on grain side and stronger capillary action occurred on the flesh side.”
  • Page 9 Line no 299, “The relative content of AFSCO in flesh side was almost higher than 60%, while it in grain side was less than 60% (shown as the blue waterfull curve in Fig.7 d).” may be written as “The relative content of AFSCO in flesh side was almost higher than 60%, while it in grain side was less than 60% (shown as the blue waterfull curve in Fig.7d).”
  • Page 9 Line no 301, “As the increase of the fatliquoring time, the contents of AFSCO both in flesh side and in grain side increased.” may be written as “With the increase of the fatliquoring time, the contents of AFSCO both in flesh side and in grain side were
  • Page 9 Line no 311, “It should be noted that the content of AFSCO in the location where was 0.2-0.3 mm to the grain side was lowest and had little change in 90 min.” may be written as “It should be noted that the content of AFSCO in the location where it was 0.2-0.3 mm to the grain side was lowest and had little change in 90 min.”
  • Page 9 Line no 314, “It can be concluded that the dosage (15%) of fatliquoring agent in this experiment was insufficient for its evenly distributing in the whole section of the leather.” may be written as “It can be concluded that the dosage (15%) of fatliquoring agent in this experiment was insufficient for its evenly distribution in the whole section of the leather.”
  • Page 9 Line no 318, “Beside of the effect of time, the dosage of fatliquoring agent was an important factor which influenced greatly on the quality of leather.” may be written as “Besides the effect of time, the dosage of fatliquoring agent was an important factor which influenced greatly on the quality of leather.”
  • Page 9 Line no 319, “Thus the distribution of fatliquoring agent with different dosages were investigated further, which fluorescence images and quantification analysis were shown in Fig. 8.” may be written as “Thus, the distribution of fatliquoring agent with different dosages were investigated further, whose fluorescence images and quantification analysis were shown in Fig. 8.”
  • Page 9 Line no 323, “And the fatliquoring agent was mainly concentrated in the flesh layer, the inner part of the crust leather had hardly fatliquoring agent.” may be written as “And the fatliquoring agent was mainly concentrated in the flesh layer, the inner part of the crust leather hardly had fatliquoring agent.”
  • Page 9 Line no 324, “As the dosage of AFSCO increased to 10%, the distribution of fatliquoring agent in leather was modified greatly.” may be written as “As the dosage of AFSCO was increased to 10%, the distribution of fatliquoring agent in leather was modified greatly.”

Conclusions

  • Page 10 Line no 336, “Based on the AF-labeled castor oil, a fluorescent tracing technique for visually investigating and quantifying the mass 338 transfer of fatliquoring agent in leather was established.” may be written as “Based on the AF-labeled castor oil, a fluorescent tracing technique, for visually investigating and quantifying the mass transfer of fatliquoring agent in leather, was established.”

Figures

  • Page 6 Figure no 2, axis scaling is not clear.
  • Page 6 Figure no 2, label the figure properly.
  • Page no 5 figure 2, what does the graph c represent?

    Additional Comments

    Either give list of abbreviations or complete name of chemicals in brackets.

    Graphs should be labeled properly.

Author Response

Dear editors and reviewers:

Thank you very much for your valuable comments on our paper submitted to Materials (materials-1562935). The manuscript has been revised according to your valuable and professional comments and we hope meet with your approval.

(1). Page 1 Line no 9, “Fatliquoring is one of the most important processes in leather making, in which the collagen fibers are split effectively while the crust acquired good softness and mechanical strength.” may be written as “Fatliquoring is one of the most important processes in leather making, in which the collagen fibers are splitted effectively while the crust acquired good softness and mechanical strength.”

Response: It has been revised.

(2). Page 1 Line no 16, “The crucial fluorescent tracer was synthesized successfully by the reaction of castor oil successively with maleic anhydride and 5-aminofluorescein, which was confirmed by FTIR, 1H NMR and DSC.” may be written as “The crucial fluorescent tracer was synthesized favorably by the reaction of castor oil successively with maleic anhydride and 5-aminofluorescein, which was confirmed by FTIR, 1H NMR and DSC.”

Response: It has been revised.

(3). Page 1 Line no 19, “And then it was applied to monitor the fatliquoring process real-timely.” may be written as “And then it was applied to monitor the fatliquoring process real-time.”

Response: It has been revised.

(4). Page 1 Line no 29, “Without the lubrication of the natural fats, the tanned collagen fibers would stick together when the water was lost and the wet blue is also easy to be cracking.” may be written as “Without the lubrication of the natural fats, the tanned collagen fibers will stick together when the water will lose and cracking of wet blue takes place easily.”

Response: According to your advice, we revised it.

(5). Page 1 Line no 36, “It should be noted that the improvement of fat-liquoring agent to the mechanical properties of leather is strongly associated with their distribution in the crust.” may be written as “It should be noted that the improvement of fat-liquoring agent in the mechanical properties of leather is strongly associated with its distribution in the crust.”

Response: It has been revised.

(6). Page 1 Line no 41, “Additionally, the chemicals combined previously with collagen fibers would influence the penetration and combination of subsequent chemicals, and a typical example is the fading effect to dye of acrylic polymers.” may be written as “Additionally, the chemicals combined previously with collagen fibers will influence the penetration and combination of subsequent chemicals, and a typical example is the fading effect to dye of acrylic polymers.”

Response: It has been revised.

(7). Page 1 Line no 43, “It is undoubtedly that the penetrating and distributing of fat-liquoring agent in the whole cross section of leather is a very complex process.” may be written as “It is undoubtedly that the penetration and distribution of fat-liquoring agent in the whole cross section of leather is a very complex process.”

Response: They have been revised.

(8). Page 2 Line no 46, “Therefore the investigation of the mass transfer processes of fat-liquoring agent in wet blue as well as its distribution is crucial to understanding its structure-activity relationship, which is very important for designing and developing of new fat-liquoring agent.” may be written as “Therefore, the investigation of the mass transfer processes of fat-liquoring agent in wet blue as well as its distribution are crucial to understand its structure-activity relationship, which is very important for designing and developing of new fat-liquoring agent.”

Response: They have been revised.

(9). Page 2 Line no 49, “However, the researches in this field were mainly focus on developing new kinds of fatliquoring agents and rarely on their mass transfer and distribution in wet blue.” may be written as “However, the researches in this field are mainly focused on developing new kinds of fatliquoring agents and rarely on their mass transfer and distribution in wet blue.”

Response: According to your advice, we revised it.

(10). Page 2 Line no 53, “But these staining methods are relatively complicated in sample preparing and the slice is easy to be broken in dying process. What more important is that these methods are difficult in monitoring the mass transfer of fat-liquoring agents in real time.” may be written as “But these staining methods are relatively complicated in sample preparation and the slice is easy to be broken in dying process. What more important is that these methods are difficult in monitoring the mass transfer of fat-liquoring agents in real-time.”

Response: They have been revised.

(11). Page 2 Line no 74, “However, few papers about investigating the mass transfer of fatliquoring agent in leather by using fluorescent tracing method were reported.” may be written as “However, few papers about investigating the mass transfer of fatliquoring agent in leather by using fluorescent tracing method, were reported.”

Response: The comma has been added.

(12). Page 2 Line no 81, “While what important for researching its mass transfer is to develop a fluorescent tracer which has similar structure with sulfated castor oil.” may be written as “While, what important for researching its mass transfer is to develop a fluorescent tracer which has similar structure with sulfated castor oil.”

Response: The comma has been added into the sentence according to your advice.

(13). Page 2 Line no 90, “Finally, the mass transfer of sulfated castor oil containing 0.5% fluorescent tracer in goat wet blue was researched in details.” may be written as “Finally, the mass transfer of sulfated castor oil containing 0.5% fluorescent tracer in goat wet blue was investigated in details.”

Response: The word “researched” has been changed into “investigated”.

(14). Page 3 Line no 109, “Then 19.6 grams of maleic anhydride was introduced into the system at 110 oC and reacted for 4 h to obtained carboxylate castor oil (CCO).” may be written as “Then 19.6 grams of maleic anhydride was introduced into the system at 110 oC and reacted for 4 to obtain carboxylate castor oil (CCO).”

Response: The word has been corrected.

(15). Page 3 Line no 112, “After the carboxylate castor oil was dissolved completely, 34.7 mg AF was used to react with the modified oil above for 114 min under the catalyzing of 19.1 mg EDC.” may be written as “After the carboxylate castor oil was dissolved completely, 34.7 mg AF was used to react with the above modified oil for 60 min under the catalyzation of 19.1 mg EDC.”

Response: The word of “above” has been brought to the front of “modified oil” and “catalyzing” has been changed into its noun form.

(16). Starting tab spacing of headings 2.1, 2.2.1 and 2.2.2 is not equal.

Response: The format has been corrected.

(17). Page 3 Line no 127, Characterizations should be characterization.

Response: This word has been revised as it singular form.

(18). Page 4 Line no 129, “The dried castor oil, CCO and AFCO was dropped on a BaF2 wafer for FTIR spectra analysis, respectively.” May be written as “The dried castor oil, CCO and AFCO was dropped on a BaF2 wafer for FTIR spectra analysis, respectively.”

Response: The word of “a” has been deleted.

(19). Page no 4 Line no 134, “About 15 mg of castor oil, CCO and AFCO was respectively dissolved in 0.5 mL of CDCl3 solvent in a 5 mm-diameter sample tube for NMR of 1H (AVANCE II 400, Bruker, Germany) analysis.” may be written as “About 15 mg of castor oil, CCO and AFCO respectively were dissolved in 0.5 mL of CDCl3 solvent in a 5 mm-diameter sample tube for NMR of 1H (AVANCE II 400, Bruker, Germany) analysis.”

Response: The grammar problem has been corrected.

(20). Page 4 Line no 138, “About 10 mg of castor oil, CCO and AFCO was respectively sealed in an aluminum pan with an empty sealed aluminum pan as the reference.” may be written as “About 10 mg of castor oil, CCO and AFCO respectively were sealed in an aluminum pan with an empty sealed aluminum pan as the reference. “

Response: The grammar problem has been corrected.

(21). Page 4 Line no 139, “The DSC curves of oil sample above were conducted on DSC250 (TA, USA) in the temperature ranged from -80 °C to 80°C with a constant heating rate of 5 °C/min under a nitrogen flow.” may be written as “The DSC curves of above oil sample were conducted on DSC250 (TA, USA) in the temperature ranged from -80 °C to 80°C with a constant heating rate of 5 °C/min under a nitrogen flow.”

Response: The word of “above” has been brought to the front of “oil sample”.

(22). Headings from 2.3.1 to 2.3.4 either capitalized each first word or not.

Response: The format of 2.3.4 was wrong and it has been corrected.

(23). Page 4 Line no 155, “200% fresh water was added into the drum and the pH of the liquid was adjusted to 3.4 by using 0.4% formic acid in 10 min.” may be written as “200% fresh water was added into the drum and the pH of the liquid was adjusted to 3.4 by using 0.4% formic acid for 10 min.”

Response: It has been corrected.

(24). Page 4 Line no 156, “And then, 4% chrome powder was added at 30oC for retanning 60 min later, it was basified by using 0.2% sodium formate to pH 4.0 in 30 min.” may be written as “And then, 4% chrome powder was added at 30oC for retanning 60 min later, it was basified by using 0.2% sodium formate to pH 4.0 for 30 min.”

Response: The word “in” has been changed into “for”.

(25). Page 4 Line no 160, “Subsequently, the leather was fat-liquored in 200% water containing with 10% AFSCO for 90 min at 50oC” may be written as “Subsequently, the leather was fat-liquored in 200% water containing with 10% AFSCO for 90 min at 50oC.”

Response: This word has been deleted.

(26). Page 4 Line no 164, “At the end of fatliquoring, the oil was fixed through decreasing the pH of liquor to 4.0 by 1.0% formic acid.” may be written as “At the end of fatliquoring, the oil was fixed by decreasing the pH of liquor to 4.0 by 1.0% formic acid.”

Response: The related word has been replaced.

(27). Page 4 Line no 174, “The slice was immersed into the Sudan IV solution above to be stained for 1min, and then washed successively by 50% (v/v) ethanol solution, 30% (v/v) ethanol solution and deionized water.” may be written as “The slice was immersed into the above-mentioned Sudan IV solution to be stained for 1min, and then washed successively by 50% (v/v) ethanol solution, 30% (v/v) ethanol solution and deionized water.”

Response: The sequence of the words has been optimized.

(28). Page 5 Line no 183, “The leather was sampled respectively at 30, 60 and 90 min, in which the dosage of fatliquoring agent was 15% and other procedures were the same as section 2.4.1.” may be written as “The leather was sampled respectively at 30, 60 and 90 min, in which the dosage of fatliquoring agent was 15% and other procedures were the same as in section 2.4.1.”

Response: The preposition “in” has been added.

(29). Page 5 Line no 185, “Furthermore, in order to research the distribution of fatliquoring agent with different dosage, 5%, 10% and 20% AFSCO was respectively used in the fatliquroing of leather.” may be written as “Furthermore, in order to research the distribution of fatliquoring agent with different dosages, 5%, 10% and 20% AFSCO were respectively used in the fatliquroing of leather.”

Response: The grammar problem has been corrected.

(30). Page 5 Line no 197, “The characteristic band of hydroxyl group at 3400 cm-1 almost disappeared and a new peak corresponding to C=C at 1641 cm-1 appeared, which proved the esterification of castor oil with maleic anhydride reacted.” may be written as “The characteristic band of hydroxyl group at 3400 cm-1 almost disappeared and a new peak corresponding to C=C at 1641 cm-1 appeared, which proved the esterification of castor oil with maleic anhydride reacted.”

Response: This word has been removed.

(31). Page 5 Line no 209, “As shown in Fig. 3A, the peaks at δ 0.69-0.85 ppm are assigned to the protons of terminal methyl groups of castor oil, while the chemical shift at 1.04-1.67 ppm attributed to the protons of methylene groups.” may be written as “As shown in Fig. 3A, the peaks at δ 0.69-0.85 ppm were assigned to the protons of terminal methyl groups of castor oil, while the chemical shift at 1.04-1.67 ppm was attributed to the protons of methylene groups.”

Response: The grammar problem has been corrected.

(32). Page 5 Line no 211, “The chemical shift between 1.91 and 2.2 ppm showed the protons of -CH2C=O groups.” may be written as “The chemical shifts between 1.91 and 2.2 ppm showed the protons of -CH2C=O groups.”

Response: The plural form of “shift” has been applied.

(33). Page 6 Line no 230, “Castor oil was a special vegetable oil with good low-temperature fluidity, and no crystallization temperature was recorded on its DSC curve, which different from other natural oil such as sunflower oil.” may be written as “Castor oil was a special vegetable oil with good low-temperature fluidity, and no crystallization temperature was recorded on its DSC curve, which was differed from other natural oil such as sunflower oil.”

Response: The passive voice has been used.

(34). Page 7 Line no 235, “After reacting with maleic anhydride, the pour point temperature of carboxylate castor oil decreased to -63.59 oC” may be written as “After reacting with maleic anhydride, the pour point temperature of carboxylate castor oil was decreased to -63.59 oC.”

The word has been added.

(35). Page 7 Line no 236, “It was because the molecular mobility as well as the unsaturation of the product increased as the grafting of maleic acid.” may be written as “It was because the molecular mobility as well as the unsaturation of the product was increased with the grafting of maleic acid.”

Response: The problem has been amended.

(36). Page 7 Line no 237, “However, the pour point temperature of AFCO increased to -47.63 oC, which might be due to the limitation of aromatic aminofluorescein on the rotation freedom of oil molecule.” may be written as “However, the pour point temperature of AFCO was increased to -47.63 oC, which might be due to the limitation of aromatic aminofluorescein on the rotation freedom of oil molecule.”

Response: It was corrected.

(37). Page 7 Line no 250, “It could be observed that the fluorescence intensity of AFCO increased with the increase of pH value.” may be written as “It could be observed that the fluorescence intensity of AFCO was increased with the increase of pH value.”

Response: It was corrected.

(38). Page 7 Line no 252, “The intensity of the emission maximum was very low at pH 6, while it increased to 70.74 and 159.85a.u.when the pH value increased to 7 and 8, respectively.” may be written as “The intensity of the emission maximum was very low at pH 6, while it was increased to 70.74 and 159. 85a.u. (spacing) when the pH value was increased to 7 and 8, respectively.”

Response: They were corrected.

(39). Page 7 Line no 254, “Furthermore, the 254 intensity of the emission maximum increased greatly to 443.87 and 599.01a.u.at pH 9 and 10, respectively.” may be written as “Furthermore, the 254 intensity of the emission maximum was increased greatly to 443.87 and 599.01a.u. (spacing) at pH 9 and 10, respectively.”

Response: They were corrected.

(40). Page 8 Line no 264, “3.5. Observation the distribution of fatliquoring agent in leather” may be written as “3.5. Observation of distribution of fatliquoring agent in leather”

Response: The “of” has been inserted.

(41). Page 8 Line no 277, “According to the change of fluorescent intensity, shown as Fig. 6c, it also revealed that the fatliquoring agent mainly distributed in the flesh layer and rarely in grain layer of the leather when its dosage was relatively insufficient (10%).” may be written as “According to the change of fluorescent intensity, shown as Fig. 6c, it was also revealed that the fatliquoring agent mainly distributed in the flesh layer and rarely in grain layer of the leather when its dosage was relatively insufficient (10%).”

Response: It has been corrected.

(42). Page 8 Line no 289, “Relying on the fluorescent tracing technology for fatliquoring agent constructed above, the mass transfers of AFSCO at different time were studied in details, which results were shown in Fig. 7a-d.” may be written as “Relying on the fluorescent tracing technology for fatliquoring agent constructed above, the mass transfers of AFSCO at different time were studied in details, whose results were shown in Fig. 7a-d.”

Response: The word “which” in this sentence has been replaced by “whose”.

(43). Page 9 Line no 293, “It reflected that the fatliquoring agent penetrated into the leather mainly from flesh side, which was because that the woven structure of collagen fibers on flesh side was much looser than it on grain side and stronger capillary action occurred on the flesh side.” may be written as “It was reflected that the fatliquoring agent penetrated into the leather mainly from flesh side, which was because of woven structure of collagen fibers on flesh side which was much looser than it on grain side and stronger capillary action occurred on the flesh side.”

Response: This sentence was modified carefully.

(44). Page 9 Line no 299, “The relative content of AFSCO in flesh side was almost higher than 60%, while it in grain side was less than 60% (shown as the blue waterfull curve in Fig.7 d).” may be written as “The relative content of AFSCO in flesh side was almost higher than 60%, while it in grain side was less than 60% (shown as the blue waterfull curve in Fig. 7d).”

Response: The spacing here has been deleted.

(45). Page 9 Line no 301, “As the increase of the fatliquoring time, the contents of AFSCO both in flesh side and in grain side increased.” may be written as “With the increase of the fatliquoring time, the contents of AFSCO both in flesh side and in grain side were

Response: They were corrected.

(46). Page 9 Line no 311, “It should be noted that the content of AFSCO in the location where was 0.2-0.3 mm to the grain side was lowest and had little change in 90 min.” may be written as “It should be noted that the content of AFSCO in the location where it was 0.2-0.3 mm to the grain side was lowest and had little change in 90 min.”

Response: The word “it” has been added.

(47). Page 9 Line no 314, “It can be concluded that the dosage (15%) of fatliquoring agent in this experiment was insufficient for its evenly distributing in the whole section of the leather.” may be written as “It can be concluded that the dosage (15%) of fatliquoring agent in this experiment was insufficient for its evenly distribution in the whole section of the leather.”

Response: This word has been revised as it noun form.

(48). Page 9 Line no 318, “Beside of the effect of time, the dosage of fatliquoring agent was an important factor which influenced greatly on the quality of leather.” may be written as “Besides the effect of time, the dosage of fatliquoring agent was an important factor which influenced greatly on the quality of leather.”

Response: The correct word has been used.

(49). Page 9 Line no 319, “Thus the distribution of fatliquoring agent with different dosages were investigated further, which fluorescence images and quantification analysis were shown in Fig. 8.” may be written as “Thus, the distribution of fatliquoring agent with different dosages were investigated further, whose fluorescence images and quantification analysis were shown in Fig. 8.”

Response: The correct word has been used.

(50). Page 9 Line no 323, “And the fatliquoring agent was mainly concentrated in the flesh layer, the inner part of the crust leather had hardly fatliquoring agent.” may be written as “And the fatliquoring agent was mainly concentrated in the flesh layer, the inner part of the crust leather hardly had fatliquoring agent.”

Response: It has been corrected.

(51). Page 9 Line no 324, “As the dosage of AFSCO increased to 10%, the distribution of fatliquoring agent in leather was modified greatly.” may be written as “As the dosage of AFSCO was increased to 10%, the distribution of fatliquoring agent in leather was modified greatly.”

Response: The passive voice has been used.

(52). Page 10 Line no 336, “Based on the AF-labeled castor oil, a fluorescent tracing technique for visually investigating and quantifying the mass transfer of fatliquoring agent in leather was established.” may be written as “Based on the AF-labeled castor oil, a fluorescent tracing technique, for visually investigating and quantifying the mass transfer of fatliquoring agent in leather, was established.”

Response: Depending on this good advice, two commas have been added.

(53). Page 6 Figure no 2, axis scaling is not clear.

Page 6 Figure no 2, label the figure properly.

Response: The resolution and label of the figure in page 6 has been improved.

(54).Page no 5 figure 2, what does the graph c represent?

Response: The graph c of figure 2 in page 5 is the FTIR spectrum of fluorescent tracer (AFCO), i.e. the product of carboxylate castor oil which was labeled by 5-aminofluorescein. A related explanation was added into the manuscript. “When the carboxylate castor oil was labeled by 5-aminofluorescein, a new peak was observed at 1590 cm-1 (shown in Fig. 2c) and it was a characteristic peak of amide II band.”

(55). Either give list of abbreviations or complete name of chemicals in brackets.

Response: Complete names of some chemicals have been supplied in brackets. Such as: “1-(3-Dimethylaminopropyl)-3-ethylcarbodiimide hydrochloride (EDC), 5-aminofluorescein (AF), Sudan IV and Nile Blue sulphate were purchased from Macklin Biochemical Co., Ltd (Shanghai, China). Goat skin in wet blue state was obtained from Shandong Juncheng Leather Co., Ltd. A technical grade basic chromium sulphate (CTS, basicity, 33%; Cr2O3 content, 25%) were supplied from Brother Enterprise Holding Co., Ltd. (Zhejiang, China), and a technical grade nonionic degreasing agent (FB, 40%) was offered by Leahou Light Industrial of New Material Co., Ltd. (Shandong, China).”

(56). Graphs should be labeled properly.

Response: The graphs have been improved.